# Delirium in patients with COVID-19 treated in the intensive care unit

**Jae Hoon Lee[1], Won Ho Han[1], June Young Chun[2], Young Ju Choi[2], Mi Ra Han[3], Jee Hee Kim[4] ***

**1** Critical Care Medicine, National Cancer Center, Goyang, South Korea, **2** Department of Internal Medicine, National Cancer Center, Goyang, South Korea, **3** Biostatistics Collaboration Team, National Cancer Center, Goyang, South Korea, **4** Department of Anesthesiology, National Cancer Center, Goyang, South Korea

* anesth@ncc.re.kr

## Abstract

Coronavirus disease 2019 (COVID-19) can lead to acute organ dysfunction, and delirium is associated with long-term cognitive impairment and a prolonged hospital stay. This retrospective single-center study aimed to investigate the risk factors for delirium in patients with COVID-19 infection receiving treatment in an intensive care unit (ICU). A total of 111 patients aged >18 years with COVID-19 pneumonia who required oxygen therapy from February 2021 to April 2022 were included. Data on patient demographics, past medical history, disease severity, delirium, and treatment strategies during hospitalization were obtained from electronic health records. Patient characteristics and risk factors for delirium were analyzed. Old age (P < 0.001), hypertension (P < 0.001), disease severity (Sequential Organ Failure Assessment score) (P < 0.001), mechanical ventilator support (P < 0.001), neuromuscular blocker use (P < 0.001), and length of stay in the ICU (P < 0.001) showed statistically significant differences on the univariable analysis. Multivariable analysis with backward selection revealed that old age (odds ratio, 1.149; 95% confidence interval, 1.037–1.273; P = 0.008), hypertension (odds ratio, 8.651; 95% confidence interval, 1.322–56.163; P = 0.024), mechanical ventilator support (odds ratio, 226.215; 95% confidence interval, 15.780–3243.330; P < 0.001), and length of stay in the ICU (odds ratio, 30.295; 95% confidence interval, 2.539–361.406; P = 0.007) were significant risk factors for delirium. In conclusion, old age, ICU stay, hypertension, mechanical ventilator support, and neuromuscular blocker use were predictive factors for delirium in COVID-19 patients in the ICU. The study findings suggest the need for predicting the occurrence of delirium in advance and preventing and treating delirium.

## Introduction

Coronavirus disease 2019 (COVID-19) is a disease that may lead to severe respiratory syndrome. Since the end of 2019, COVID-19 has rapidly spread, eventually becoming a global pandemic. Until 2022, numerous institutions conducted studies on treatments for the acute and chronic phases of this disease and its complications in people of all ages and sexes [1–5].

**Data Availability Statement:** All relevant data are within the manuscript and its Supporting Information files.

**Funding:** Initials of the authors who received each award: WHH Grant numbers awarded to each

author: NCC 2212490-2 The full name of each funder: National Cancer Center, Korea URL of each funder website: https://www.ncc.re.kr/indexEn.ncc The funders had no role in study design, data collection and analysis, decision to publish, or preparation of the manuscript.

**Competing interests:** The authors have declared that no competing interests exist.

Among these complications, delirium, which is characterized by a change in consciousness that diminishes the ability to concentrate and maintain attention [6–8], can occur over a short period of time in patients with COVID-19. In the intensive care unit (ICU), delirium is primarily nonspecific, preventable, and reversible. Nevertheless, managing and treating delirium following its occurrence are complex, with delirium causing long-term cognitive impairment and being associated with a prolonged hospital stay. Recognizing preventable factors and preventing them may reduce the mortality rate and shorten the hospitalization period [9, 10]. Previous studies on delirium have mainly included patients without COVID-19. In contrast, studies involving COVID-19 patients have mainly focused on complications in the acute phase, such as acute respiratory distress syndrome (ARDS) [1–3, 11] and cardiovascular or inflammatory complications [12–16]. Additionally, only a few studies on neurological manifestations have been reported.

Therefore, the present study aimed to identify risk factors associated with delirium development and to investigate preventable factors in patients with COVID-19 receiving treatment in the ICU.

## Materials and methods

This retrospective single-center study was conducted from February 2021 to April 2022. It included a total of 111 patients aged >18 years who had COVID-19, as confirmed by a positive reverse transcription–polymerase chain reaction test result, who required oxygen therapy (nasal prong, $O_2$ mask, high-flow nasal cannula, mechanical ventilator support, extracorporeal membrane oxygenation [ECMO]), and who were admitted to the ICU. Delirium was defined as a Richmond Agitation and Sedation Scale equivalent score of -3 or higher according to the Confusion Assessment Method for the Intensive Care Unit and Diagnostic and Statistical Manual of Mental Disorders (Fifth Edition) criteria, as evaluated by an attending physician in patients responding to verbal stimulation [6–10]. ICU stay was defined as a stay in the ICU for >2 weeks. The patients' characteristics included sex, age, smoking status, body mass index, factors for the assessment of underlying medical conditions and severity (Acute Physiology and Chronic Health Evaluation II [APACHE II] score and Sequential Organ Failure Assessment [SOFA] score), medications (steroids, tocilizumab, neuromuscular [NM] blocker, analgesic, sedative agent, and low-molecular-weight heparin), oxygen therapy (mechanical ventilator support, continuous renal replacement therapy, ECMO, and prone position), and length of stay in the ICU. The relationship between the occurrence of delirium and related factors was analyzed.

All patients were treated with remdesivir, antibiotics, steroids, anticoagulants, and oxygen therapy in accordance with the ARDS treatment and COVID guidelines [5, 17]. Depending on the patient's compliance, an NM blocker (vecuronium at 1–3 mg/hour), sedative agents (dexmedetomidine at 0.2–1 mcg/kg/hour and midazolam at 0.02–0.1 mg/kg/hour), and analgesic agents (propofol at 5–50 mcg/kg/min and remifentanil at 0.02–0.2 mcg/kg/min) were administered. Nonpharmacologic management (orientation protocol, cognitive stimulation, facilitation of physiologic sleep, early mobilization, minimized use of physical restraints, and visual and hearing aids) was also carried out in all patients. If an attending physician observed no improvement in patients with delirium after pharmacologic treatment (quetiapine, dexmedetomidine, haloperidol, and lorazepam), treatment was performed after consulting a psychiatrist at the hospital [18–20].

This study was conducted in accordance with the principles embodied in the Declaration of Helsinki and was approved by the Institutional Review Board of the National Cancer Center (approval no.: NCC 2022–0322). The data access period for the manuscript is from October 24, 2022, the date of IRB approval, to December 31, 2022. The requirement for the acquisition of informed consent from patients was waived owing to the retrospective nature of this study.

## Statistical analysis

Continuous variables are expressed as mean ± standard deviation or median (minimum–maximum). Categorical variables were compared between groups using Pearson's chi-squared test or Fisher's exact test. In contrast, continuous variables were compared using the t-test for parametric data and the Wilcoxon rank-sum test for nonparametric data. Multivariable analysis with backward selection, which included variables with P < 0.1 in the univariable analysis, was conducted. All statistical analyses were performed using SAS for Windows version 9.4 (SAS Institute, Cary, NC, USA), with statistical significance set at P < 0.05.

## Results

### Patient characteristics

Overall, 111 patients (average age: 64.1 years) were treated for COVID-19 in the ICU. Of these patients, four died due to pneumonia exacerbation (n = 2), multi-organ failure (n = 1), and multiple stomach cancer metastases (n = 1) after discharge from the ICU. Among the 111 patients, 32 patients (10 patients without delirium vs. 22 patients with delirium) required ventilator support; out of these 32 patients, 8 patients (3 patients without delirium vs. 5 patients with delirium) received ECMO treatment due to ARDS exacerbation. Additionally, 19 patients (8 patients without delirium vs. 11 patients with delirium) were treated with an NM blocker to optimize the conditions for endotracheal intubation, mechanical ventilation, and prone positioning, whereas 32 patients (10 patients without delirium vs. 22 patients with delirium) required analgesic and sedative agents. Furthermore, 26 patients (10 patients without delirium vs. 16 patients with delirium) stayed in the ICU for >2 weeks. Among 26 patients with delirium (23.4%), 16 received medications from an attending physician, whereas 10 patients who were irritable or did not respond to medications were managed through consultation with a psychiatrist (Table 1). For patients with sleep problems caused by delirium, 6.25–25 mg of quetiapine was administered. For irritable patients, 2.5–5 mg of haloperidol was administered intramuscularly if no improvement was observed. For patients with insomnia, 0.125–0.25 mg of triazolam was added. For patients with anxiety, 0.25–1.2 mg of alprazolam was administered.

### Analysis of risk factors for delirium

Univariable analysis of patients with COVID-19 and delirium indicated significant differences with respect to old age (odds ratio [OR], 1.076; 95% confidence interval [CI], 1.029–1.124); P < 0.001), hypertension (HTN) (OR, 6.300; 95% CI, 2.166–18.321; P < 0.001), disease severity (SOFA score) (OR, 1.612; 95% CI, 1.248–2.083; P < 0.001), mechanical ventilator support (OR, 41.247; 95% CI, 11.781–144.412; P < 0.001), NM blocker use (OR, 7.058; 95% CI, 2.432–20.485; P < 0.001), and length of stay in the ICU (OR, 10.762; 95% CI, 3.910–29.625; P < 0.001) (Table 2). Multivariable analysis with backward selection revealed that old age (OR, 1.149; 95% CI, 1.037–1.273; P = 0.008), HTN (OR, 8.651; 95% CI, 1.332–56.163; P = 0.024), mechanical ventilator support (OR, 226.215; 95% CI, 15.780–3243.330; P < 0.001), NM blocker use (OR, 0.037; 95% CI, 0.002–0.765; P<0.033) and length of stay in the ICU (OR, 30.295; 95% CI, 2.539–361.406; P = 0.007) were risk factors for delirium (Table 3).

### Characteristics of patients with mechanical ventilator support

Comparison of risk factors for delirium according to the presence of delirium in patients with mechanical ventilator support identified HTN only as a significant factor (20% vs. 81.8%,

**Table 1. Patient characteristics.**

|  | Patients (N = 111) | Percentage |
|---|---|---|
| **Sex** |  |  |
| Male | 59 | 53.2 |
| Female | 52 | 46.9 |
| Age (years) | 64.1 ± 13.1 |  |
| Body mass index (kg/m$^2$) | 25.1 ± 4.2 |  |
| **Underlying disease** |  |  |
| Diabetes mellitus | 29 | 26.1 |
| Hypertension | 55 | 49.6 |
| Chronic obstructive pulmonary disease | 6 | 5.4 |
| Cardiovascular disease | 17 | 15.3 |
| Chronic kidney disease | 4 | 3.6 |
| Dementia | 5 | 4.5 |
| Stroke | 3 | 2.7 |
| Depression | 4 | 3.6 |
| Smoking | 34 | 30.6 |
| From diagnosis to ICU admission | 5.4 (0.0–25.0) |  |
| APACHE II score | 27 ± 6.6 |  |
| SOFA score | 4.3 ± 2.1 |  |
| **Steroid therapy** |  |  |
| Standard dose | 65 | 58.6 |
| High dose | 46 | 41.4 |
| LMWH | 91 | 82.0 |
| Tocilizumab | 24 | 21.6 |
| Mechanical ventilator support | 32 | 28.8 |
| CRRT | 5 | 4.5 |
| ECMO | 8 | 7.2 |
| Prone position | 17 | 15.3 |
| NM blocker | 19 | 17.1 |
| Analgesic agents | 32 | 28.8 |
| Sedative agents | 32 | 28.8 |
| **ICU stay duration** |  |  |
| <2 weeks | 85 | 76.6 |
| >2 weeks | 26 | 23.4 |
| **Delirium** | 26 | 23 |
| **Outcome** |  |  |
| Survival | 107 | 96.4 |
| Death | 4 | 3.6 |

APACHE II, Acute Physiology and Chronic Health Evaluation II; SOFA, Sequential Organ Failure Assessment; LMWH, low-molecular-weight heparin; CRRT, continuous renal replacement therapy; ECMO, extracorporeal membrane oxygenation; NM, neuromuscular.

P = 0.002). Age, NM blocker use, and length of stay in the ICU did not significantly differ between the groups (Table 4).

## Discussion

Delirium is associated with an increased duration of hospitalization, including ICU stay, and with increased complication and mortality rates in critically ill patients [21–29]. Delirium

**Table 2. Comparison between patients without delirium and those with delirium.**

| | Without delirium (N = 85) (%) | With delirium (N = 26) (%) | P value |
|---|---|---|---|
| **Sex** | | | |
| Male | 44 (51.8) | 15 (57.7) | 0.596 |
| Female | 41 (48.2) | 11 (42.3) | |
| Age (years) | 61.8 ± 13.2 | 71.8 ± 9.2 | <0.001 |
| Body mass index (kg/m$^2$) | 25.3 ± 4.3 | 24.2 ± 3.7 | 0.237 |
| **Underlying disease** | | | |
| Diabetes mellitus | 22 (25.9) | 7 (26.9) | 0.916 |
| Hypertension | 34 (40.0) | 21 (80.8) | <0.001 |
| Chronic obstructive pulmonary disease | 4 (4.7) | 2 (7.7) | 0.623 |
| Cardiovascular disease | 10 (11.8) | 7 (26.9) | 0.115 |
| Chronic kidney disease | 2 (2.4) | 2 (7.7) | 0.233 |
| Dementia | 2 (2.4) | 3 (11.5) | 0.083 |
| Stroke | 1 (1.2) | 2 (7.7) | 0.136 |
| Depression | 1 (1.2) | 3 (11.5) | 0.039 |
| Smoking | 27 (31.8) | 7 (26.9) | 0.639 |
| From diagnosis to ICU admission | 5.8 (0–25) | 4.7 (0–23) | 0.241 |
| APACHE II score | 26 (12–55) | 31 (24–42) | <0.001 |
| SOFA score | 4 (1–13) | 5 (3–11) | <0.001 |
| **Steroid therapy** | | | |
| Standard dose | 54 (63.5) | 11 (42.3) | 0.055 |
| High dose | 31 (36.5) | 15 (57.7) | |
| LMWH | 69 (81.2) | 22 (84.6) | 0.779 |
| Tocilizumab | 16 (18.8) | 8 (30.8) | 0.195 |
| Mechanical ventilator support | 10 (11.8) | 22 (84.6) | <0.001 |
| CRRT | 2 (2.4) | 3 (11.5) | 0.083 |
| ECMO | 3 (3.5) | 5 (19.2) | 0.017 |
| Prone position | 8 (9.4) | 9 (34.6) | 0.004 |
| NM blocker | 8 (9.4) | 11 (42.3) | <0.001 |
| Analgesic agents | 10 (11.8) | 22 (84.6) | <0.001 |
| Sedative agents | 10 (11.8) | 22 (84.6) | <0.001 |
| ICU stay (>2 weeks) | 10 (11.8) | 16 (61.5) | <0.001 |

APACHE II, Acute Physiology and Chronic Health Evaluation II score; SOFA, Sequential Organ Failure Assessment; LMWH, low-molecular-weight heparin; CRRT, continuous renal replacement therapy; ECMO, extracorporeal membrane oxygenation; NM, neuromuscular.

occurs in 14–55% of inpatients in general wards and 20–80% of patients in ICUs, depending on ventilator use, sedation depth, immobility, patient characteristics, and disease severity [20, 30, 31].

Risk factors for severe COVID-19 include age >65 years and comorbidities (e.g., cancer, chronic disease, obesity, immunosuppression), which are also regarded as risk factors commonly associated with delirium. Nonetheless, only a few studies have investigated the relationship between COVID-19 and delirium [9, 18, 32]. The present study analyzed the predictive risk factors for delirium in patients with COVID-19 and identified statistically significant differences in age, HTN, mechanical ventilator support, NM blocker use, and length of stay in the ICU among patients with delirium.

Previous studies reported old age, underlying diseases, APACHE II score, SOFA score, hospital stay, and smoking as known risk factors for delirium [7, 19]. In the present study,

**Table 3. Multivariable analysis of factors associated with delirium.**

| | Univariable analysis | | Multivariable analysis | |
|---|---|---|---|---|
| | OR (95% CI) | P value | OR (95% CI) | P value |
| Sex | 0.787 (0.324–1.910) | 0.597 | | |
| Age (years) | 1.076 (1.029–1.124) | <0.001 | 1.149 (1.037–1.273) | 0.008 |
| Body mass index (kg/m²) | 0.936 (0.838–1.044) | 0.235 | | |
| **Underlying disease** | | | | |
| Diabetes mellitus | 1.055 (0.391–2.849) | 0.916 | | |
| Hypertension | 6.300 (2.166–18.321) | <0.001 | 8.651 (1.322–56.163) | 0.024 |
| Chronic obstructive pulmonary disease | 1.688 (0.291–9.788) | 0.559 | | |
| Cardiovascular disease | 2.763 (0.930–8.212) | 0.067 | | |
| Chronic kidney disease | 3.458 (0.462–25.860) | 0.227 | | |
| Dementia | 5.412 (0.853–34.347) | 0.073 | | |
| Stroke | 6.994 (0.608–80.453) | 0.119 | | |
| Depression | 10.956 (1.088–110.347) | 0.042 | | |
| Smoking | 0.791 (0.297–2.108) | 0.640 | | |
| Time from diagnosis to ICU admission | 0.967 (0.883–1.059) | 0.471 | | |
| SOFA | 1.612 (1.248–2.083) | <0.001 | | |
| **Steroid therapy** | | | | |
| High dose | 2.375 (0.971–5.811) | 0.058 | | |
| LMWH | 1.275 (0.386–4.218) | 0.690 | | |
| Tocilizumab | 1.917 (0.709–5.182) | 0.200 | | |
| Mechanical ventilator support | 41.247 (11.781–144.412) | <0.001 | 226.215 (15.780–3243.330) | <0.001 |
| CRRT | 5.412 (0.853–34.347) | 0.073 | | |
| ECMO | 6.506 (1.438–29.435) | 0.015 | | |
| Prone position | 5.096 (1.717–15.119) | 0.003 | | |
| NM blocker | 7.058 (2.432–20.485) | <0.001 | 0.037 (0.002–0.765) | 0.033 |
| Sedative agents | 41.247 (11.781–144.412) | <0.001 | | |
| ICU stay (>2 weeks) | 10.762 (3.910–29.625) | <0.001 | 30.295 (2.539–361.406) | 0.007 |

OR, odds ratio; CI, confidence interval; SOFA, Sequential Organ Failure Assessment; LMWH, low-molecular-weight heparin; CRRT, continuous renal replacement therapy; ECMO, extracorporeal membrane oxygenation; NM, neuromuscular.

delirium occurred in 26 patients (26.4%) during their ICU stay, which is similar to the incidence of delirium (19–50%) reported by previous studies conducted at other institutions in patients with and without COVID-19 [32–35]. Consistent with the findings of previous studies, mechanical ventilator support, old age, length of stay in the ICU, and underlying diseases were identified as risk factors for delirium in this study. Several patients with ARDS and respiratory distress requiring ventilator support needed deep sedation, which may elucidate why sedation agent administration and NM blocker use were risk factors. Furthermore, a high incidence of delirium was observed mainly in patients with HTN. Similarly, previous reports comparing the presence of delirium among patients receiving ventilator support showed that HTN was the only risk factor [31, 36–39].

In patients requiring mechanical ventilators, dexmedetomidine is known to cause less delirium than benzodiazepines [40]. In this study, benzodiazepine and dexmedetomidine were used as sedatives. In this study, 81.82% of patients used benzodiazepine as a sedative, and 90.91% used dexmedetomidine. Although benzodiazepines were used in relatively high doses per hour in patients with delirium, there was no statistical correlation (1.3±1.3 vs. 2.5±3.3, P = 0.142). Relatively small doses of dexmedetomidine were used in patients with delirium,

**Table 4. Factors associated with delirium in patients receiving ventilator support.**

| | Without delirium (N = 10) (%) | With delirium (N = 22) (%) | P value |
|---|---|---|---|
| **Sex** | | | |
| Male | 4 (40.0) | 13 (59.1) | 0.450 |
| Female | 6 (60.0) | 9 (40.9) | |
| Age (years) | 62.3 ± 14.2 | 69.8 ± 7.1 | 0.141 |
| Body mass index (kg/m$^2$) | 25.5 ± 3.1 | 24.8 ± 3.3 | 0.576 |
| **Underlying disease** | | | |
| Diabetes mellitus | 2 (20.0) | 7 (31.8) | 0.681 |
| Hypertension | 2 (20.0) | 18 (81.8) | 0.002 |
| Chronic obstructive pulmonary disease | 1 (10.0) | 2 (9.1) | 1.000 |
| Cardiovascular disease | 2 (20.0) | 6 (27.3) | 1.000 |
| Chronic kidney disease | 1 (10.0) | 2 (9.1) | 1.000 |
| APACHE II score | 33.5 (18–55) | 31 (25–42) | 0.554 |
| SOFA score | 7 (5–13) | 5.5 (4–11) | 0.183 |
| High-dose steroid | 8 (80.0) | 14 (63.6) | 0.440 |
| CRRT | 2 (20.0) | 3 (13.6) | 0.637 |
| ECMO | 3 (30.0) | 5 (22.7) | 0.681 |
| Prone position | 2 (20.0) | 8 (36.4) | 0.440 |
| NM blocker | 8 (80.0) | 11 (50.0) | 0.141 |
| Duration (days) | 3.1±3.5 | 3.2±4.9 | 0.962 |
| **Sedative agents** | | | |
| Dexmedetomidine | 9 (90.0) | 20 (90.9) | 1.000 |
| Dose (mcg/kg/hr) | 0.3±0.2 | 0.2±0.2 | 0.388 |
| Duration (days) | 5.9±5.1 | 6.6±9.7 | 0.824 |
| Midazolam | 6 (60.0) | 18 (81.8) | 0.218 |
| Dose (mg/hr) | 1.3±1.3 | 2.5±3.3 | 0.142 |
| Duration (days) | 3.7±4.9 | 7.4±9.8 | 0.165 |
| ICU stay (>2 weeks) | 3 (30.0) | 15 (68.2) | 0.062 |

APACHE II, Acute Physiology and Chronic Health Evaluation II; SOFA, Sequential Organ Failure Assessment; CRRT, continuous renal replacement therapy; ECMO, extracorporeal membrane oxygenation; NM, neuromuscular.

but there was no statistical correlation (0.3±0.2 vs. 0.2±0.2, P = 0.388). Patients requiring ventilator treatment owing to COVID-19 infection often require relatively high doses of sedatives to maintain sedation [41]. This study also required a high dose to maintain sedation; therefore, a combination of benzodiazepine and dexmedetomidine was often necessary. Thus, unlike previous studies, there was no difference in the occurrence of delirium depending on the type of sedative agents.

The present study has some limitations, particularly its limited sample size and retrospective single-center study design. While our results confirmed that sedative agent administration and NM blocker use were risk factors for delirium, the dose and selection of sedative agents, as well as the duration of sedative agent use, were not considered; therefore, their effects on delirium require further evaluation. Additionally, data on the diagnosis of delirium were collected from medical records, which might lead to a potential bias. Finally, whether delirium had improved in patients was impossible to ascertain because the patients were not followed up after discharge from the ICU.

Despite these limitations, this study identified the following risk factors for delirium in patients receiving ICU treatment for COVID-19 infection: old age, ICU stay, hypertension,

mechanical ventilator support, and NM blockers. Thus, when an elderly COVID patient with hypertension is admitted to the ICU, the amount of sedative agent and NM blocker used to maintain ventilator care should be minimized, and a protocol should be established for ventilator-weaning to prevent delirium (e.g., daily spontaneous breathing trials). [42]. Additional multicenter studies with large patient populations should analyze the risk factors to establish plans for preventing and treating delirium.

## Conclusions

Old age, ICU stay, HTN, mechanical ventilator support, and NM blocker use were predictive factors for delirium in patients with COVID-19 infection receiving ICU treatment. The study findings suggest the need for predicting the occurrence of delirium in advance and preventing and treating delirium.

## Supporting information

**S1 Checklist. STROBE statement—checklist of items that should be included in reports of observational studies.**
(DOCX)

**S1 Data.**
(XLSX)

## Author Contributions

**Conceptualization:** June Young Chun, Young Ju Choi, Jee Hee Kim.

**Data curation:** Jae Hoon Lee, Mi Ra Han.

**Formal analysis:** Jae Hoon Lee, Won Ho Han, Mi Ra Han.

**Methodology:** Jae Hoon Lee, Won Ho Han, Young Ju Choi.

**Writing – original draft:** Jae Hoon Lee, Won Ho Han.

**Writing – review & editing:** Jae Hoon Lee, Won Ho Han, Jee Hee Kim.

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
