## [Decision Letter · Decision Letter 0]

29 Aug 2023

PONE-D-23-21282Delirium in patients with COVID-19 treated in the intensive care unitPLOS ONE

Dear Dr. Kim,

Thank you for submitting your manuscript to PLOS ONE. After careful consideration, we feel that it has merit but does not fully meet PLOS ONE’s publication criteria as it currently stands. Therefore, we invite you to submit a revised version of the manuscript that addresses the points raised during the review process.

We look forward to receiving your revised manuscript.

Kind regards,

Chiara Lazzeri

Academic Editor

PLOS ONE

Additional Editor Comments:

The paper is well written. However we suggest to add a paragraph describilig the clinical implications of the main findings. Morevoer, some information on dosages and type of sedatives should be added

An English mother-tongue expert should revise the manuscript.

Reviewers' comments:

Reviewer's Responses to Questions

**Comments to the Author**

1. Is the manuscript technically sound, and do the data support the conclusions?

Reviewer #1: Partly

2. Has the statistical analysis been performed appropriately and rigorously? 

Reviewer #1: Yes

3. Have the authors made all data underlying the findings in their manuscript fully available?

Reviewer #1: Yes

4. Is the manuscript presented in an intelligible fashion and written in standard English?

Reviewer #1: Yes

5. Review Comments to the Author

Reviewer #1: I commend the authors for their efforts in conducting a study that investigates the risk factors for delirium in COVID-19 patients within the intensive care unit.

The manuscript presents a retrospective single-center study investigating risk factors for delirium in COVID-19 patients receiving treatment in an intensive care unit (ICU). The study's objective is to identify predictive factors for the occurrence of delirium, and it analyzes various patient characteristics and medical interventions. Overall, the manuscript is technically sound and presents data that support its conclusions. However, there are several strengths and areas for improvement to consider.

Strengths:

Clearly Defined Methods: The manuscript provides clear and detailed descriptions of the study design, data collection methods, and statistical analysis. This transparency enhances the manuscript's credibility.

Ethical Considerations: The study's adherence to ethical principles, including IRB approval and waiver of informed consent due to the retrospective nature, is commendable.

Relevance: The study addresses an important clinical question related to COVID-19 patients in the ICU, which is a relevant topic.

Acknowledgment of Limitations: The manuscript clearly discusses the study's limitations, such as the small sample size, single-center design, and retrospective nature. This demonstrates the authors' awareness of potential biases and limitations.

Statistical Analysis: The use of multivariable analysis to identify risk factors is appropriate and strengthens the study's conclusions.

Areas for Improvement:

Data Interpretation: While the study identifies risk factors for delirium, it would be beneficial to provide more in-depth interpretation of the clinical implications of these findings. How can this knowledge be applied in clinical practice to prevent and manage delirium in COVID-19 patients?

Discussion of Sedative Agents: The manuscript mentions sedative agent administration as a risk factor for delirium. Further discussion on the types and dosages of sedative agents used, as well as their potential impact on delirium, would enhance the discussion section.

Follow-up Data: The manuscript mentions that follow-up data after discharge from the ICU were not available. Discussing the importance of follow-up and potential directions for future research in this area could add depth to the discussion.

Clarity of Figures and Tables: Ensure that figures and tables are clear and well-labeled for readers to easily interpret the data presented.

Language and Grammar: While the manuscript is generally well-written, a final proofreading pass for language and grammar errors would enhance its professionalism.

Minor Comments about the Language

The manuscript is presented in a generally intelligible fashion and is written in standard English. The language used is clear, correct, and mostly unambiguous. However, there are a few instances where clarity could be improved:

In the abstract, the sentence "Old age, intensive care unit stay, hypertension, mechanical ventilator support, and neuromuscular blocker use were predictive factors for the occurrence of delirium in patients with coronavirus disease 2019 infection receiving treatment in the intensive care unit" is quite long and could be broken down into shorter, more concise sentences for improved readability

Results Section:

Original lengthy sentence: "Among these complications, delirium, which is characterized by a change in consciousness that diminishes the ability to concentrate and maintain attention [1–3], can occur over a short period of time in patients with COVID-19."

Revised: "Among these complications, delirium can occur rapidly in COVID-19 patients. It is characterized by changes in consciousness, affecting concentration and attention [1–3]."

Original lengthy sentence: "Furthermore, 26 patients (10 patients without delirium vs. 16 patients with delirium) stayed in the ICU for >2 weeks."

Revised: "Additionally, 26 patients, of which 10 had no delirium and 16 had delirium, had ICU stays exceeding 2 weeks."

Discussion Section:

Original complex sentence: "While our results confirmed that sedative agent administration and NM blocker use were risk factors for delirium, the dose and selection of sedative agents, as well as the duration of sedative agent use, were not considered; therefore, their effects on delirium require further evaluation."

Revised: "Although our results confirmed that sedative agent administration and NM blocker use were risk factors for delirium, we did not assess the specific doses, types, or durations of sedative agents. Further evaluation is needed to understand their effects on delirium."

Original complex sentence: "Delirium occurs in 14–55% of inpatients in general wards and 20–80% of patients in ICUs, depending on ventilator use, sedation depth, immobility, patient characteristics, and disease severity."

Revised: "The occurrence of delirium varies widely, with rates ranging from 14% to 55% in general wards and 20% to 80% in ICUs. Factors influencing this variation include ventilator use, sedation depth, immobility, patient characteristics, and disease severity."

These revisions aim to make the sentences more concise and easier to understand, enhancing the overall flow and comprehension of the manuscript.

Conclusion:

The manuscript presents a technically sound study investigating risk factors for delirium in COVID-19 patients in the ICU. It adheres to ethical principles and discusses its limitations transparently. To further strengthen the manuscript, it would be beneficial to provide a more in-depth interpretation of the clinical implications of the findings and to address the areas for improvement highlighted above.

6. PLOS authors have the option to publish the peer review history of their article (what does this mean?). If published, this will include your full peer review and any attached files.

Reviewer #1: No

---

## [Author Response · Author response to Decision Letter 0]

11 Sep 2023

Response to Reviewer’s Comments

Reviewer(s)' Comments to Author:

There is lack of evidence to support the development of X. Why choose N as sample size? The author should add baseline analysis of different type of SRBC to adjust the differences. 

1. Data Interpretation: While the study identifies risk factors for delirium, it would be beneficial to provide more in-depth interpretation of the clinical implications of these findings. How can this knowledge be applied in clinical practice to prevent and manage delirium in COVID-19 patients?

Reply:

We would like to thank the Reviewer for the time and effort spent reviewing our manuscript and for providing comments, which have helped us improve our manuscript. We have made revisions based on your comments and have provided our point-by-point responses below. We hope that our responses and revisions appropriately address your comments.

A description has been added after the Study limitations paragraph of the discussion section (please see page 18).

This study identified the following risk factors for delirium in patients receiving ICU treatment for COVID-19 infection: old age, ICU stay, hypertension, mechanical ventilator support, and NM blockers. Thus, when an elderly COVID patient with hypertension is admitted to the ICU, the amount of sedative agent and NM blocker used to maintain ventilator care should be minimized, and a protocol should be established for ventilator-weaning to prevent delirium (e.g., daily spontaneous breathing trials).

2. Discussion of Sedative Agents: The manuscript mentions sedative agent administration as a risk factor for delirium. Further discussion on the types and dosages of sedative agents used, as well as their potential impact on delirium, would enhance the discussion section.

Reply:

Based on your comment, a description has been added to the study limitations paragraph of the discussion section (please see page 18).

In this study, 81.82% of patients used benzodiazepine as a sedative, and 90.91% used dexmedetomidine. Although benzodiazepines were used in relatively high doses per hour in patients with delirium, there was no statistical correlation (1.3±1.3 vs. 2.5±3.3, P=0.142). Relatively small doses of dexmedetomidine were used in patients with delirium, but there was no statistical correlation (0.3±0.2 vs. 0.2±0.2, P=0.388). Patients requiring ventilator treatment owing to COVID-19 infection often require relatively high doses of sedatives to maintain sedation [41]. This study also required a high dose to maintain sedation; therefore, a combination of benzodiazepine and dexmedetomidine was often necessary. Thus, unlike previous studies, there was no difference in the occurrence of delirium depending on the type of sedative agents.

3. Follow-up Data: The manuscript mentions that follow-up data after discharge from the ICU were not available. Discussing the importance of follow-up and potential directions for future research in this area could add depth to the discussion.

Reply:

Unfortunately, in the early stages of COVID-19 infection in Korea, patients were divided into those who were severe cases requiring oxygen treatment and those who did not require this treatment due a mild presentation. Treatment for the severe group was then referred to hospitals that had the capacity to do so. Once patients’ severe symptoms were cured, they were immediately transferred to a specialized hospital that treats mild symptoms. This was a single-center study; therefore, there was no follow-up of patients at the end of treatment. Although some patients could be followed up at the hospital where the study was conducted, most continued their remaining treatment at another hospital after transferring.

We completely agree with the reviewer that further research would help track patients treated at our hospital.

4. Clarity of Figures and Tables: Ensure that figures and tables are clear and well-labeled for readers to easily interpret the data presented.

Reply:

According to your comments, we have made some changes to ensure that the subsections for the categories are more distinguishable using bold writing, making the information clearer to the reader. We have removed any blank/empty cells and have formatted the tables to ensure that the text is presented in a clear manner.

5. Language and Grammar: While the manuscript is generally well-written, a final proofreading pass for language and grammar errors would enhance its professionalism.

Reply:

The manuscript has now undergone a final proofreading as per your suggestion.

6. Minor Comments about the Language

In the abstract, the sentence "Old age, intensive care unit stay, hypertension, mechanical ventilator support, and neuromuscular blocker use were predictive factors for the occurrence of delirium in patients with coronavirus disease 2019 infection receiving treatment in the intensive care unit" is quite long and could be broken down into shorter, more concise sentences for improved readability

Reply:

We have revised the sentence as per the reviewer’s suggestion to enhance conciseness, clarity, and overall readability (please see page 2):

In conclusion, old age, ICU stay, hypertension, mechanical ventilator support, and neuromuscular blocker use were predictive factors for delirium in COVID-19 patients in the ICU.

---

## [Editor Report · Decision Letter 1]

18 Sep 2023

Delirium in patients with COVID-19 treated in the intensive care unit

PONE-D-23-21282R1

Dear Dr. Kim,

We’re pleased to inform you that your manuscript has been judged scientifically suitable for publication and will be formally accepted for publication once it meets all outstanding technical requirements.

Kind regards,

Chiara Lazzeri

Academic Editor

PLOS ONE
---

## [Editor Report · Acceptance letter]

4 Nov 2023

PONE-D-23-21282R1 

Delirium in patients with COVID-19 treated in the intensive care unit 

Dear Dr. Kim:

I'm pleased to inform you that your manuscript has been deemed suitable for publication in PLOS ONE. Congratulations! Your manuscript is now with our production department. 

Kind regards, 

on behalf of

Dr. Chiara Lazzeri 

Academic Editor

PLOS ONE